# Federated Black-Box Adaptation for Semantic Segmentation

**Jay N. Paranjape**
Dept. of Electrical and Computer Engineering
The Johns Hopkins University
Baltimore, USA
jparanj1@jhu.edu

**Shameema Sikder**
Wilmer Eye Institute
The Johns Hopkins University
Baltimore, USA

**S. Swaroop Vedula**
Malone Center for Engineering in Healthcare
The Johns Hopkins University
Baltimore, USA

**Vishal M. Patel**
Dept. of Electrical and Computer Engineering
The Johns Hopkins University
Baltimore, USA

## Abstract

Federated Learning (FL) is a form of distributed learning that allows multiple institutions or clients to collaboratively learn a global model to solve a task. This allows the model to utilize the information from every institute while preserving data privacy. However, recent studies show that the promise of protecting the privacy of data is not upheld by existing methods and that it is possible to recreate the training data from the different institutions. This is done by utilizing gradients transferred between the clients and the global server during training or by knowing the model architecture at the client end. In this paper, we propose a federated learning framework for semantic segmentation without knowing the model architecture nor transferring gradients between the client and the server, thus enabling better privacy preservation. We propose *BlackFed* - a black-box adaptation of neural networks that utilizes zero order optimization (ZOO) to update the client model weights and first order optimization (FOO) to update the server weights. We evaluate our approach on several computer vision and medical imaging datasets to demonstrate its effectiveness. To the best of our knowledge, this work is one of the first works in employing federated learning for segmentation, devoid of gradients or model information exchange. Code: https://github.com/JayParanjape/blackfed/tree/master

## 1   Introduction

With data-driven methods becoming immensely popular in Artificial Intelligence (AI) research and applications, there has been a surge in data collection and curation across the world. This has, in turn, led to the development of AI models that require substantial amounts of data to train. Federated Learning (FL) [34] was developed as a viable approach towards training such models by effectively harnessing the data collected at different centers across the world. Through FL, it becomes possible for multiple institutions to collaborate and build a joint model that learns from all of them, while reducing the burden of collecting more data individually. However, collaborations between different institutions present a non-trivial challenge due to disparity in data distributions as well as the imperative of safeguarding data privacy. Consequently, FL aims to jointly learn a shared model that performs well on data from all participating institutions without exchanging raw data. Various FL approaches have been proposed in the literature [1, 10, 12, 16, 25, 22, 35, 37, 41, 44, 55, 56, 30, 34, 29, 31]. Among

38th Conference on Neural Information Processing Systems (NeurIPS 2024).

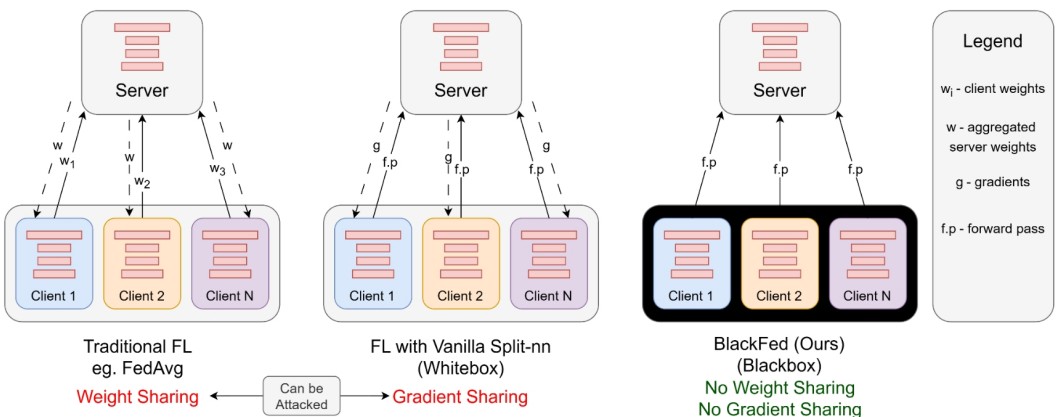

Figure 1: Comparison of our method against traditional FL methods. Existing FL methods are primarily "white-box" as they involve transfer of model weights [34], or gradients[20]. In contrast, our method only utilizes forward passes to update the client and does not require sharing weights or gradients, making it a "black-box" model.

these, FedAvg [34] was one of the pioneering FL methods, which proposes training local models at every center using local data and periodically averaging the local model weights to craft a global server model. Various subsequent works improve FedAvg by improving the local model updates [30, 31] or global updates [24, 29].

While FL was primarily proposed for classification tasks, it is also suited for other computer vision tasks such as segmentation. Generating annotations in segmentation entails creating pixel-level annotations per image, that are more tedious to label than classification tasks. Consequently, it is not always possible for a single institution to collect a large amount of data, underscoring the importance of collaborative efforts. A few approaches have been proposed in the literature for FL-based segmentation [36, 5, 18, 43, 53, 14]. However, all these methods for segmentation and classification, while not involving raw data transfer, employ techniques like model information transfer [34, 30, 36] or gradient transfer [24, 20], as shown in Figure 1. However, recent research has revealed that these techniques are vulnerable to attacks that can recreate training data from the participating centers, thus undermining the privacy preserving characteristic of FL [11, 54, 17, 57, 27, 26, 58, 15, 23]. These attacks employ methods like gradient inversion [54, 17, 57] or adapting model architecture and weights [11]. In this work, we propose a new approach, named BlackFed, for segmentation using FL that does not involve gradient transfer between the server and the client and at the same time, passes no knowledge about the client model architecture to the server, thereby avoiding the necessary conditions for these attacks, as shown in Figure 1. This is done by formulating the FL learning problem as a distributed learning problem using split neural networks (split-nn) [20] and combining first order and zero order optimization techniques for training. Our contributions are as follows:

1. We introduce BlackFed - a black-box algorithm that facilitates distributed learning for semantic segmentation without transferring model information or gradients between the client and the server. For this, we formulate the FL problem using split-nn and use first and zero order optimization for training the server and the clients, respectively.

2. We suggest a method to reduce the effect of catastrophic forgetting in BlackFed by retaining client-wise checkpoints in the server.

3. We evaluate the proposed approach on four segmentation datasets and show its effectiveness as a distributed learning method by showing improvements over individual training.

## 2   Related Work

**Federated Learning for Segmentation:**   FL for the segmentation tasks was motivated by its immense application in the medical domain. Consequently, various methods were introduced in this field [33, 6, 13, 19, 28, 32, 42, 49, 50, 52, 38]. For instance, [32] attempts to learn a model for brain

tumor segmentation by utilizing data from multiple institutions. FedSM [38] attempts to mitigate the effect of non-iid nature of the data from different centers on the global server model. However, most of the approaches for medical segmentation focus on the problem of segmenting out the foreground from the background (one class problem). There are relatively fewer works in the literature for multi-class segmentation [14, 36, 5, 18, 43, 53]. FedSeg [36] deals with the class label inconsistency problem that may be present at the local clients. In other words, it builds a robust system that works well even when the clients have annotations for only a subset of the classes. FedDrive [14], on the other hand, sets up various benchmarks for FL algorithms on multi-class datasets like Cityscapes [9]. However, all the existing algorithms require the global server to know the model architecture used in the clients and thus, these methods are vulnerable to recent attacks [11]. In our work, we develop an algorithm to perform multi-class segmentation which does not require gradients or client model sharing.

**Split Neural Networks:** Split networks [20] were introduced as an alternative to FedAvg-like techniques which require model sharing. They offer an approach to perform collaborative learning by splitting a larger network into two segments. The latter segment of the network is shared across all centers and placed at the global server, while the former part is distributed such that each center possesses its own sub-network. During training, the clients perform a forward pass using their data and send the encoded features to the server, which further passes the features to the subsequent layers in the network. The server and client models are trained using backpropagation, where the gradients from the first layer of the server model are sent back to each of the clients. Split networks have mainly been used in the literature for the task of classification, mostly in the medical domain [20, 40, 47]. Our work marks one of the initial explorations of split learning in the context of semantic segmentation. Furthermore, split networks are shown to be more robust to reconstruction attacks which arise from sharing model information [48]. Nonetheless, they remain susceptible to gradient leakage attacks since there is gradient transfer between the server and the client. In this work, we formulate the problem of distributed semantic segmentation using split network and introduce a training algorithm that does not require gradient-sharing.

## 3 Black-box Adaptation

In the following section, we define the problem of semantic segmentation under the federated setting. Then, we describe our proposed formulation for the black-box setting and the algorithm.

### 3.1 Preliminaries

The setting of FL consists of $N$ clients, denoted by $\mathbb{C}_i$, where $i \in 1, 2, ..., N$ and a global server $\mathbb{G}$. Each of the clients has a local dataset, consisting of images $X_i = \left\{ x_j^i \in \mathbb{R}^{H \times W \times C}; j \in \{1, 2, ..., n^i\} \right\}$ and their corresponding ground truth segmentation maps $Y_i = \left\{ y_j^i \in \mathbb{R}^{H \times W \times N_c}; j \in \{1, 2, ..., n^i\} \right\}$. Here, $x_j^i$ and $y_j^i$ represent the $j^{th}$ image and its corresponding segmentation mask, while $n^i$ represents the number of data points in client $i$ and $N_c$ represents the number of classes in the output. Note that the distributions of the input images vary among the clients. Each client uses its dataset to learn a function $f^i : \mathbb{R}^{H \times W \times C} \to \mathbb{R}^{H \times W \times N_c}$, which is parameterized by $\Theta^i$, that minimizes its local loss function as follows:

$$\underset{\Theta^i}{\arg\min} \, \mathbb{L}_i = \frac{1}{n^i} \sum_{j=1}^{n^i} l(f(x_j^i; \Theta^i), y_j^i), \tag{1}$$

where $l$ denotes the loss per data point. The global server, on the other hand, tries to aggregate the knowledge of all the clients by fusing the parameters of the individual clients to optimize a joint loss function as follows:

$$\underset{\Theta^i}{\arg\min} \, \mathbb{L} = \frac{1}{N} \sum_{i=1}^{N} \mathbb{L}_i. \tag{2}$$

A commonly used algorithm, called FedAvg [34], proposes a linear combination of client parameters to optimize the loss function, defined in Eq. 2, as follows:

$$\Theta^i \leftarrow \sum_{i=1}^{N} \frac{n^i}{\sum_{i=1}^{N} n^i} \Theta^i. \tag{3}$$

However, this formulation requires the server to be aware of the exact model architecture or function of the client, which can results in data leakage [11].

## 3.2 Proposed Algorithm

**Proposed Problem Formulation:** In this work, we model the problem of distributed learning using a split neural network, which takes away the requirement of the server being aware of the client architecture. In this case, each client learns a function $f^i : \mathbb{R}^{H \times W \times C} \to \mathbb{R}^{H' \times W' \times C'}$, which is parameterized by $\Theta^i$. Similarly, the global server learns a function $g : \mathbb{R}^{H' \times W' \times C'} \to \mathbb{R}^{H \times W \times N_c}$, which is parameterized by $\Phi$. Thus, the forward pass for a given center is given as follows:

$$\hat{y}_j^i = g(f(x_j^i; \Theta^i); \Phi), \tag{4}$$

where $\hat{y}_j^i$ denotes the predicted segmentation map. Hence, the objective function of a given client is as follows:

$$\underset{\Theta^i, \Phi}{\arg\min} \, \mathbb{L}^i, i \in \{1, 2, ..., N\} = \frac{1}{n^i} \sum_{j=1}^{n^i} l(\hat{y}_j^i, y_j^i; \Theta^i, \Phi). \tag{5}$$

As in the existing FL literature, the goal of our approach is that after training, all clients should benefit from each other. Hence, during evaluation, given any client, we aim to perform well on data from other clients as well as its own data, thus showing good generalization. More specifically, we want to optimize any combination of data and client as follows:

$$\min \mathbb{L}^{ik}, i \in \{1, 2, ..., N\}, k \in \{1, 2, ..., N\} = \frac{1}{n^i} \sum_{j=1}^{n^i} l(\hat{y}_j^i, y_j^i; \Theta^k, \Phi). \tag{6}$$

One way to optimize this equation is by attending to every client in a round-robin fashion. This involves selecting a client, performing the forward pass, as defined in Eq. 4 and then performing backpropagation to update the server and client using the client loss function defined in Eq. 5. Performing this operation several times in a round-robin fashion enables the server to learn from all client sources, and hence, improves the overall performance. We term this method as "White-box Round-Robin FL", and is similar to the existing methods in the literature [24, 20]. However, recent works have shown that such a method which involves transfer of gradients between the server and client can be utilized to regenerate the training data, thus undermining the privacy preservation principle of FL [54, 23]. Hence, we add one more constraint - i.e. no gradient can flow back from the server to the client in Eq. 5.

**BlackFed Algorithm:** To optimize the clients without using gradients, we utilize a ZOO method called Simultaneous Perturbation Stochastic Approximation with Gradient Correction (SPSA-GC) [39] which involves perturbing the weights of the client model slightly and approximating a two-sided gradient based on the change in the loss function due to the perturbations. However, this method was developed to perform black-box adaptation of pretrained foundation models and hence, expects the server model to be initialized with good weights, which is not the case in our formulation, making it non-trivial. To overcome this, we propose to iteratively use an alternating optimization technique, which factorizes the optimization problem in Eq. 5 into two optimization problems as follows:

$$\underset{\Theta^i}{\arg\min} \, \mathbb{L}^i, i \in \{1, 2, ..., n\} = \frac{1}{n^i} \sum_{j=1}^{n^i} l(\hat{y}_j^i, y_j^i; \Theta^i | \Phi),$$

$$\underset{\Phi}{\arg\min} \, \mathbb{L}^i, i \in \{1, 2, ..., n\} = \frac{1}{n^i} \sum_{j=1}^{n^i} l(\hat{y}_j^i, y_j^i; \Phi | \Theta^i). \tag{7}$$

During training, we first select a client using the round-robin policy. Next, keeping the server weights fixed, we train the client weights using SPSA-GC for a few iterations. Next, we fix the client weights and use a first order optimizer (i.e. Adam-W) to optimize the server for a few iterations. This process is repeated multiple times. During inference for a client, we simply run the forward pass as described in Eq. 4 using the final weights of the client and server. The training process is described in Algorithm 1. We refer to this approach as BlackFed v1.

**Algorithm 1** Proposed Algorithm for BlackFed v1

---

**Input:** (i) N, number of clients
   (ii) c_e, number of epochs to train one client
   (iii) s_e, number of epochs to train server
   (iv) runs, number of complete round-robin runs $n \geq 0$
**Output:** (i) $\Phi$, server model weights
   (ii) $\Theta$, array of client model weights

   **BlackFed (N, c_e, s_e, runs) Begin:**
   initialize $\Phi, \Theta$
   **for** r = 1,2,...,runs **do**
      **for** i = 1,2,...,N **do**
         **for** j = 1,2,...,c_e **do**
            $\Theta[i] \leftarrow$ SPSA-GC$(\Theta[i])$   //Zero order optimization for client
         **end for**
         **for** j = 1,2,...,s_e **do**
            $\Phi \leftarrow \Phi - \eta_s \frac{\partial \mathbb{L}^i}{\partial \Phi}$   //First order optimization for server, where $\eta_s$ is the learning rate
         **end for**
      **end for**
   **end for**
   **Return:** $\Phi, \Theta$
   **End**

---

**Reducing the Effect of Catastrophic Forgetting:** Since the model at the server is shared among all the clients and is updated in a round-robin fashion, it may happen that training with the data from a given client may cause it to unlearn patterns for the previous client. This phenomenon is often called catastrophic forgetting. This effect is observed in BlackFed v1 especially when the number of clients increases or if there is a significant change in the data distribution among clients. This causes the algorithm to perform well on certain clients and poorly on the rest of the clients. To reduce the effect of catastrophic forgetting, we propose a simple additional step during training with Algorithm 1. After updating the server weights for a given client during training, we store the updated weights of the server model in a hashmap indexed by the index of the client. During inference for a given client, we use the latest weights of the client model and the indexed weights of the server model to perform the forward pass. Note that the server state is loaded from the hashmap only during inference. During training, the server still benefits from the data from all clients and updates its weights as well as the hashmap. This approach is visualized in Figure 2. We refer to this method as BlackFed v2.

## 4 Experiments and Results

**Datasets:** For evaluating our method, we consider four publicly available datasets, namely (i) Cityscapes [9] (ii) CAMVID [4], (iii) ISIC [21, 8, 45] and (iv) Polypgen [2]. Cityscapes and CAMVID are two road-view semantic segmentation datasets with 19 and 32 classes of interest respectively, collected from multiple cities. In the FL setup, we consider each of the cities as separate clients. While CAMVID has predefined train, test and validation splits, for Cityscapes, we divide the data from each client into training, validation and testing splits for that client in a 60:20:20 ratio. In this manner, we generate 18 clients for Cityscapes and 4 clients for CAMVID. Further details about the dataset size in each center is provided in the supplementary document. The ISIC dataset corresponds to a skin lesion segmentation challenge. We generate three clients for this dataset, which utilize the datasets from ISIC 2016 [21], ISIC 2017 [8] and ISIC 2018 [45], respectively. The data every year is collected from different centers and hence, has different distribution amongst centers. Finally, PolypGen is a colon polyp segmentation dataset which was collected from six different centers. The training, validation and testing splits for each of these were done in a 60:20:20 ratio for Polypgen whereas they were already provided for ISIC. We visualize the pixel-intensity histograms of the CAMVID and ISIC datasets to verify different data distributions amongst clients in Figure 3.

**Experimental Setup:** We use a two-layer convolutional network for modeling $f$ in the client and DeepLabv3 [7] for modeling $g$ in the server. The number of output channels from the client is kept

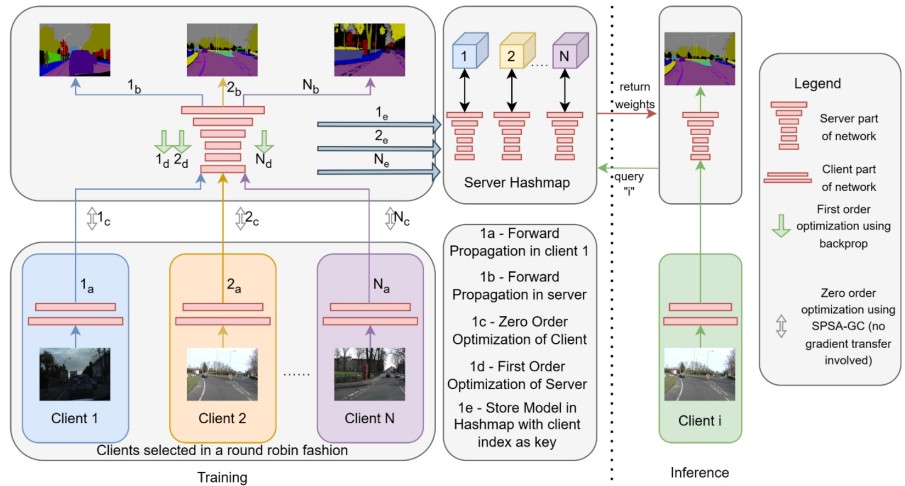

Figure 2: The BlackFed v2 Algorithm. During training, the client is selected in a round-robin fashion. Then (a) client performs a forward pass using its part of the network (b) Server performs a forward pass using its part of the network (c) With server weights fixed, client weights updated using ZOO (d) Keeping client weights fixed, server weights updated using FOO (e) The best server weights are stored in the hashmap corresponding to client index. During inference, the client performs a forward pass and calls the server with the output. Server queries the hashmap using the client index and gets its set of weights, using which the prediction is obtained. Note that there is no gradient transfer, thus making this a black-box setup.

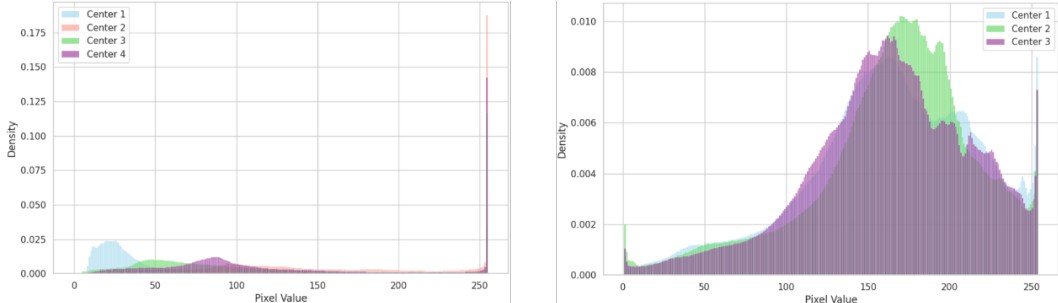

Figure 3: Pixel Density distribution of (L) the CAMVID Dataset and (R) the ISIC Dataset. Since majority of ISIC pixels are either 0 or 255 for all centers, these have been omitted for better visualization. Since each of the clients has a different distribution, data from one client can be considered as Out-of-Distribution (OOD) for other clients.

as 64. Consequently, we start the DeepLabv3 network in the server from the second layer, which expects a 64-channel input. During training, we use $c\_e = 10$ and $s\_e = 10$. The server is optimized using an Adam optimizer and the client is optimized using SPSA-GC. The learning rates of both the client and the server are set to $10^{-4}$, based on validation set performance. The batch size for all experiments is 8, and all images undergo random brightness perturbation with brightness parameter set as 2. The images for Cityscapes and CAMVID are resized to $256 \times 512$ to maintain their aspect ratio, whereas the images for ISIC and Polypgen are resized to $256 \times 256$. All experiments are done using a single Nvidia RTX A5000 GPU per client and a single Nvidia RTX A5000 GPU at the server.

**Results:** Given the trained client models and the trained server model, we assess a given client's performance with test datasets from its own local data repository and present the mIoU in the "Local" column. This metric represents the local performance of the model on its own dataset. In addition, we assess each client on test data from other centers and note down the average mIoU scores in the Out-of-Distribution ("OOD") column. This metric serves as an estimate of the general performance of a given client post-training. We consider the latter more important as it can be considered to be the direct outcome of the FL paradigm. These results are presented in Table 1 for CAMVID and Cityscapes, and in Table 2 for ISIC and Polypgen. For both tables, in the first row, each client is

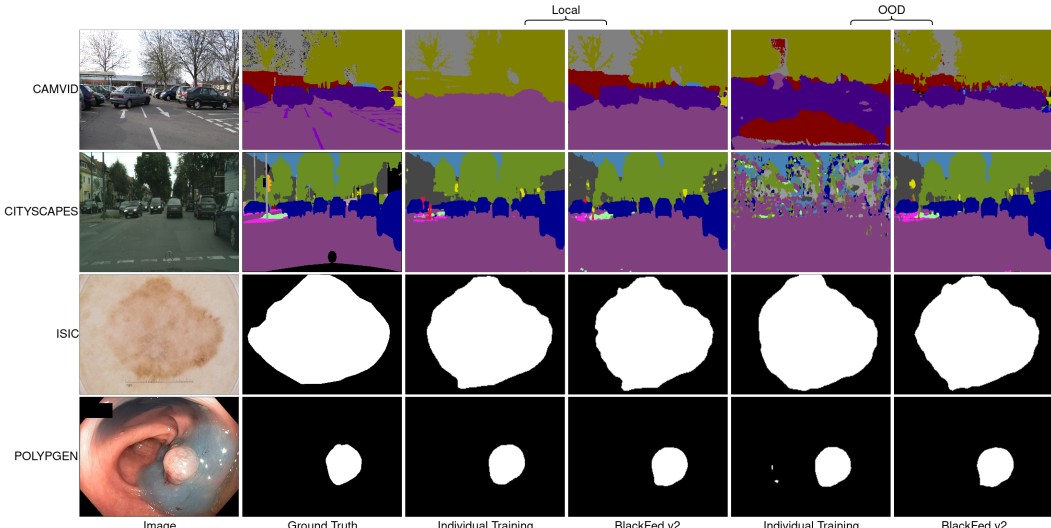

Figure 4: Comparison of our method against individual training. The third and fourth columns denote testing with the local test data, while the fifth and sixth columns denote OOD testing. Our method improves OOD performance of clients without harming their local performance.

trained on its own dataset, independently of others, indicating no collaboration. For this case, while the client performance on its own test set is high, its general performance suffers. The next two rows represent our method (BlackFed v1 and v2). Notably, BlackFed v2 generally exhibits better performance than v1 since it addresses the catastrophic forgetting that occurs during training. The following three rows represent the expected upper bounds to our performance. "Combined Training" represents the scenario where raw data can be freely shared and a single model is trained using the combined data from all clients. "White-box training" represents the case where gradients can flow freely between the server and the client. Thus, instead of the ZOO optimization, we use FOO to optimize the client part of the model. Finally, the last three rows represent the performance of traditional FL using FedAvg, and recent methods like FedPer and FedSeg, where model sharing is allowed. Here, all the clients follow the same model architecture as the server (DeepLabv3) and the server can aggregate weights from the client models. All three methods represent a relaxation over the constraints imposed in our approach, thus acting as an upper bound to the black-box performance. During training, the client and server models are trained for the same number of epochs per client. This produces a more uniform distribution of results across centers, in contrast with traditional FL methods like FedAvg, where the aggregation of weights is weighed by the client dataset size. We observe that for the Polypgen dataset, BlackFed v1 and v2 perform slightly better or on par with individual client performance for OOD case. For this case, there is little difference between the performance of v1 and v2. This behavior may be related to the data distribution of Polypgen and suggests that BlackFedv2 is not able to correctly avoid the catastrophic forgetting for centers C5 and C6. However, for rest of the scenarios, we see that v2 significantly outperforms v1 as well as the individual training cases on OOD mIoU metric. At the same time, the performance on local data does not suffer significantly as compared to the individual training. Moreover, BlackFed v2 performs on par with "Combined" and "White-box" Training. All results of BlackFed have a p-value less than 0.001, showing the statistical significance of our black-box approach. The visual comparisons for our method with the individual training is given in Fig. 4. As can be seen in all four rows, individual training can lead to overfitting, which harms the general OOD performance. Using our method, we are able to improve the OOD results across all datasets without significant decrease in the local results.

**Additional Model Architectures** While we evaluate a DeepLabv3-based server in our main experiments, we show the effectiveness of our method on additional segmentation architectures. More specifically, for the CAMVID dataset, we choose UNext [46] and SegFormer [51] as the server models and present average mIoU results across the test datasets from each client in Table 3. As can be seen from the table, using our approach can improve the performance over individual training for all three models. As the model complexity increases from UNext to DeepLab to Segformer, we

Table 1: mIoU scores for BlackFed v1 and v2 in comparison with individual and FL-based training strategies for natural datasets. "Local" represents test data from the center. "OOD" represents mean mIoU on test data from rest of the centers. For FedAvg and Combined Training, just one model is trained. Hence, its performance is noted only in each of the local test datasets. For Cityscapes, we only present the average local and OOD performance across centers for brevity. The supplementary contains an expanded version for Cityscapes.

| | CAMVID | | | | | | | | Cityscapes Average across 18 Centers | |
| | C1 | | C2 | | C3 | | C4 | | | |
| Method | Local | OOD | Local | OOD | Local | OOD | Local | OOD | Local | OOD |
|---|---|---|---|---|---|---|---|---|---|---|
| Individual | 0.63 | 0.46 | **0.83** | 0.48 | **0.85** | 0.65 | **0.79** | 0.64 | 0.50 | 0.44 |
| BlackFed v1 | 0.55 | 0.67 | 0.79 | 0.65 | 0.78 | 0.68 | 0.66 | 0.66 | 0.71 | 0.71 |
| BlackFed v2 | **0.70** | **0.72** | 0.78 | **0.66** | 0.81 | **0.70** | 0.75 | **0.72** | **0.75** | **0.74** |
| Combined Training | 0.74 | - | 0.81 | - | 0.84 | - | 0.77 | - | 0.77 | - |
| White-box Training | 0.67 | 0.73 | 0.81 | 0.72 | 0.80 | 0.68 | 0.74 | 0.70 | 0.75 | 0.75 |
| FedAvg [34] | 0.64 | - | 0.76 | - | 0.84 | - | 0.76 | - | 0.79 | - |
| FedSeg [36] | 0.71 | - | 0.79 | - | 0.83 | - | 0.77 | - | 0.81 | - |
| FedPer [3] | 0.65 | 0.57 | 0.77 | 0.68 | 0.82 | 0.71 | 0.76 | 0.66 | 0.78 | 0.72 |

Table 2: mIoU scores for BlackFed v1 and v2 in comparison with individual and FL-based training strategies for medical datasets. "Local" represents test data from the center. "OOD" represents mean mIoU on test data from rest of the centers. For FedAvg and Combined Training, just one model is trained. Hence, its performance is noted only in each of the local test datasets.

| | Polypgen | | | | | | | | | | | | ISIC | | | | | |
| | C1 | | C2 | | C3 | | C4 | | C5 | | C6 | | C1 | | C2 | | C3 | |
| Method | Local | OOD | Local | OOD | Local | OOD | Local | OOD | Local | OOD | Local | OOD | Local | OOD | Local | OOD | Local | OOD |
|---|---|---|---|---|---|---|---|---|---|---|---|---|---|---|---|---|---|---|
| Individual | 0.59 | 0.47 | **0.73** | 0.47 | **0.75** | **0.59** | 0.47 | 0.37 | **0.45** | 0.44 | **0.52** | **0.48** | 0.86 | 0.79 | 0.85 | 0.76 | 0.76 | 0.84 |
| BlackFed v1 | 0.59 | 0.55 | 0.63 | **0.51** | 0.64 | 0.56 | **0.55** | 0.47 | 0.34 | **0.53** | 0.28 | 0.40 | 0.84 | **0.82** | **0.89** | 0.80 | 0.5 | 0.69 |
| BlackFed v2 | **0.59** | **0.56** | 0.64 | 0.49 | 0.65 | 0.51 | 0.50 | **0.47** | 0.35 | 0.49 | 0.26 | 0.41 | **0.86** | 0.80 | 0.88 | **0.80** | **0.77** | **0.88** |
| Combined Training | 0.71 | - | 0.79 | - | 0.81 | - | 0.67 | - | 0.57 | - | 0.60 | - | 0.87 | - | 0.89 | - | 0.76 | - |
| White-box Training | 0.60 | 64 | 0.77 | 0.59 | 0.72 | 0.58 | 0.61 | 0.57 | 0.48 | 0.62 | 0.59 | 0.62 | 0.56 | 0.70 | 0.73 | 0.66 | 0.77 | 0.68 |
| FedAvg [34] | 0.68 | - | 0.78 | - | 0.83 | - | 0.61 | - | 0.54 | - | 0.64 | - | 0.87 | - | 0.87 | - | 0.78 | - |

observe a decrease in individual training performance. This trend is reversed for the combined case where there is more training data. This observation indicates overfitting in the individual case due to less individual training data. Using our method, we are able to correctly match the performance and trend of combined training.

## 5 Ablation Studies

**Analysis of the Order of Optimization:** In the alternating optimization proposed in Eq. 7, we choose to first update the client followed by the server. This order is important since it allows us to store the correct server weights in the hashmap. If the server is trained before the client, we found that SPSA-GC often gives unstable results and reduces the metric on the validation set after an epoch. This is corrected only when its the turn of the same client again. Conversely, in the case where the client is updated first, the server adapts in a stable manner to the client weights since backpropagation is allowed in the server. Consequently, in the same epoch, we get a higher and more stable performance

Table 3: mIoU for CAMVID dataset with varying model architectures of the global server.

| | DeepLabv3 | | | | Segformer | | | | UNext | | | |
| Method | C1 | C2 | C3 | C4 | C1 | C2 | C3 | C4 | C1 | C2 | C3 | C4 |
|---|---|---|---|---|---|---|---|---|---|---|---|---|
| Individual | 0.50 | 0.57 | 0.69 | 0.67 | 0.27 | 0.36 | 0.50 | 0.34 | 0.36 | 0.49 | **0.60** | 0.51 |
| BlackFed v2 | **0.72** | **0.69** | **0.73** | **0.72** | **0.71** | **0.69** | **0.73** | **0.72** | **0.61** | **0.53** | 0.43 | **0.56** |
| Combined Training | 0.74 | 0.81 | 0.84 | 0.77 | 0.66 | 0.77 | 0.77 | 0.72 | 0.59 | 0.67 | 0.73 | 0.67 |
| White-box Training | 0.72 | 0.76 | 0.84 | 0.76 | 0.54 | 0.55 | 0.58 | 0.57 | 0.36 | 0.49 | 0.60 | 0.51 |

Table 4: Comparison of client and server-side GFLOPS for different algorithms.

| Method | DeepLabv3 GFLOPS | | Segformer GFLOPS | | UNext GFLOPS | |
| | Client | Server | Client | Server | Client | Server |
|---|---|---|---|---|---|---|
| Centerwise | 353.6 | - | 7.5 | - | 475.39 | - |
| FedAvg [34] | 353.6 | 353.6 | 7.5 | 7.5 | 475.39 | 475.39 |
| Ours | 26.6 | 326.98 | 5.12 | 2.38 | 53.13 | 422.26 |

Table 5: Average MIoU scores for different training strategies of BlackFed. Optimizing the client followed by the server improves performance, which is further improved by maintaining the server-side hashmap.

| Method | CAMVID Aveg across 3 centers | ISIC Average across 4 centers | CityScapes Avg across 18 centers | PolypGen Avg across 6 centers |
|---|---|---|---|---|
| Optimize Server, then client | 0.67 | 0.66 | 0.53 | 0.43 |
| Optimize Client, then Server (BlackFed v1) | 0.67 | 0.76 | 0.53 | 0.50 |
| Optimize Client, then Server (BlackFed v2) | **0.72** | **0.83** | **0.74** | **0.50** |

for each client, which can be saved in the hashmap for usage during inference. We demonstrate this empirically by comparing the performance of the two training strategies in Table 5.

**Analysis of Computational Cost:** For each of the three model architectures, namely DeepLabV3, UNext and Segformer, we calculate the floating point operations required in a single forward pass at the institution end. These are shown in Table 4. If the clients were to train individual models with their local data, they would require running the entire forward pass on their local systems. This is also the case in existing FL approaches like FedAvg. In the proposed approach, as can be seen from Row 3 in the table, the client has a significantly reduced load, with the majority of computation being offloaded to the server. The server uses the respective architectures starting from the second layer, while all the clients use a two-layer convolution network, with the second convolutional layer being similar to the first layer of the respective architecture.

**Effect of Client and Server rounds:** The BlackFed algorithm takes in two hyperparameters, namely $c\_e$ and $s\_e$, that determine the number of epochs for training the client and server, respectively. We conduct experiments for observing the effect of changing these parameters on the model performance. We use the CAMVID dataset with server architecture being DeepLabv3 and note down the average mIoU score in Table 6 for different values of $c\_e$ and $s\_e$ in $\{10, 20, 30, 40, 50\}$. Interestingly, we observe that there was no significant difference in performance by increasing the number of server epochs or client epochs. While increasing the server epochs can improve training, it also makes the server more specific to a given client, thus reducing average performance. On the other hand, more

Table 6: Average mIoU scores for BlackFed v1 and v2 in comparison with individual and FL-based training strategies.

| Number of Server Epochs | Number of Client Epochs | | | | |
| | 10 | 20 | 30 | 40 | 50 |
|---|---|---|---|---|---|
| 10 | **0.71** | 0.69 | 0.68 | 0.69 | 0.68 |
| 20 | 0.70 | 0.69 | 0.69 | 0.69 | 0.69 |
| 30 | 0.65 | 0.68 | 0.70 | 0.70 | 0.66 |
| 40 | 0.71 | 0.70 | 0.68 | 0.70 | 0.70 |
| 50 | 0.67 | 0.69 | 0.70 | 0.66 | 0.69 |

ZOO-based iterations marginally improve the model. In this context, ZOO primarily serves to guide the server towards a more robust minima, resilient to perturbations in client features. Consequently, a greater number of client epochs does not notably impact model performance.

## 6 Conclusion

In this work, we introduce BlackFed, an FL algorithm that enables distributed learning without transfer of gradients or model weights. This characteristic distinguishes our approach as a black-box model compared to existing FL methods, which can be considered as white-box approaches. Recent research on attacking FL methods require the knowledge of either the gradients or the model information, thus rendering BlackFed more resistant to such attacks since gradients or weights are not shared. BlackFed consists of a global server and multiple clients, each possessing their own data. The server is optimized using first order optimization while the client weights are updated using zero order optimization in a round-robin fashion. This introduces the effect of catastrophic forgetting in the network, for which we propose a simple hashmap-based approach. During training, we store the client weights per server

so that they can be utilized during inference. With these modifications, our approach demonstrates superior results compared to non-collaborative training and matches the performance of white-box methods, despite being a black-box method itself. Extensive experimentation on the natural and medical domain datasets highlights the effectiveness of BlackFed. Through this endeavor, we aim to propel research towards the development of better privacy preserving federated learning systems. Potential directions for future research can include analysis of other policies for selecting client order during training, and its relation with the disparity in data distribution. One more interesting direction for future research would be the effect of adversarial attacks using generative models on this method. Since masks are shared between the client and server, it would be interesting to check if existing methods are able to recreate client data using them.

## Acknowledgments and Disclosure of Funding

This research was supported by a grant from the National Institutes of Health, USA; R01EY033065. The content is solely the responsibility of the authors and does not necessarily represent the official views of the National Institutes of Health. The authors have no competing interests in the paper.

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

Table 7: Data counts for CAMVID, ISIC and Polypgen datasets

| Split | CAMVID | | | | ISIC | | | Polypgen | | | | | |
| | C1 | C2 | C3 | C4 | C1 | C2 | C3 | C1 | C2 | C3 | C4 | C5 | C6 |
|---|---|---|---|---|---|---|---|---|---|---|---|---|---|
| Train | 24 | 61 | 181 | 103 | 6009 | 720 | 2000 | 153 | 180 | 274 | 136 | 124 | 52 |
| Val | 8 | 16 | 50 | 26 | 2003 | 180 | 150 | 51 | 60 | 91 | 45 | 42 | 18 |
| Test | 92 | 24 | 74 | 42 | 2003 | 379 | 600 | 52 | 61 | 92 | 46 | 42 | 18 |

Table 8: Data counts for Cityscapes dataset

| Split | Cityscapes | | | | | | | | | | | | | | | | | |
| | C1 | C2 | C3 | C4 | C5 | C6 | C7 | C8 | C9 | C10 | C11 | C12 | C13 | C14 | C15 | C16 | C17 | C18 |
|---|---|---|---|---|---|---|---|---|---|---|---|---|---|---|---|---|---|---|
| Train | 121 | 67 | 221 | 107 | 59 | 154 | 76 | 173 | 137 | 83 | 69 | 65 | 255 | 137 | 100 | 66 | 99 | 85 |
| Val | 27 | 15 | 48 | 24 | 13 | 34 | 17 | 38 | 30 | 18 | 15 | 15 | 55 | 30 | 22 | 15 | 22 | 19 |
| Test | 26 | 14 | 47 | 23 | 13 | 33 | 16 | 37 | 29 | 18 | 15 | 14 | 55 | 29 | 22 | 14 | 21 | 18 |

## A    Centerwise Dataset Information

In this section, we describe the number of data points in the training, validation and testing splits of each center of each dataset. The center represents the client in FL, where each center is in possession of data that cannot be shared directly with the other centers or to the public. CAMVID [4] has 4 centers, which are described in Table 7. Cityscapes [9] has 18 centers, which are described in Table 8. These centers represent the different cities from which the data is collected for these datasets. ISIC [8, 21, 45] has 3 centers and Polypgen [2]has 4 centers, which represent the different hospitals from which the data is collected. These results are present in Table 7.

## B    Centerwise Performance for Cityscapes Dataset

While the average mIoU is presented in the main paper, we also present the centerwise performance of DeepLabv3-based server for the Cityscapes dataset in Tables 9 and 10. It can be seen from the tables that individual training performs poorly, the reason being that each of the centers has limited amount of data. In contrast, our method makes use of data from all clients to improve performance. Here, we also see that BlackFed v2 performs better than v1 in all cases, thus showing the effectiveness of the server hashmap in countering catastrophic forgetting. With our approach, the performance of the model reaches close to the white-box methods, but without sharing any gradients or model information.

Table 9: mIoU scores for BlackFed v1 and v2 in comparison with individual and FL-based training strategies for Cityscapes. "Local" represents test data from the center. "OOD" represents mean mIoU on test data from rest of the centers. For FedAvg and Combined Training, just one model is trained. Hence, its performance is noted only in each of the local test datasets.

| Method | C1 | | C2 | | C3 | | C4 | | C5 | | C6 | | C7 | | C8 | | C9 | |
| | Local | OOD | Local | OOD | Local | OOD | Local | OOD | Local | OOD | Local | OOD | Local | OOD | Local | OOD | Local | OOD |
|---|---|---|---|---|---|---|---|---|---|---|---|---|---|---|---|---|---|---|
| Individual | 0.61 | 0.57 | 0.61 | 0.58 | 0.64 | 0.52 | 0.55 | 0.50 | 0.35 | 0.39 | 0.39 | 0.39 | 0.51 | 0.39 | 0.33 | 0.39 | 0.55 | 0.52 |
| BlackFed v1 | 0.76 | 0.72 | 0.67 | 0.71 | 0.77 | 0.71 | 0.75 | 0.71 | 0.66 | 0.70 | 0.74 | 0.71 | 0.71 | 0.71 | 0.66 | 0.72 | 0.70 | 0.70 |
| BlackFed v2 | **0.78** | **0.74** | **0.71** | **0.74** | **0.81** | **0.74** | **0.77** | **0.74** | **0.70** | **0.73** | **0.76** | **0.74** | **0.75** | **0.73** | **0.66** | **0.75** | **0.71** | **0.72** |
| Combined Training | 0.82 | - | 0.71 | - | 0.82 | - | 0.78 | - | 0.74 | - | 0.78 | - | 0.76 | - | 0.67 | - | 0.74 | - |
| White-box Training | 0.79 | 0.74 | 0.71 | 0.75 | 0.81 | 0.75 | 0.78 | 0.75 | 0.70 | 0.74 | 0.78 | 0.75 | 0.74 | 0.74 | 0.66 | 0.76 | 0.73 | 0.76 |
| FedAvg [34] | 0.85 | - | 0.75 | - | 0.84 | - | 0.80 | - | 0.74 | - | 0.80 | - | 0.78 | - | 0.70 | - | 0.76 | - |

Table 10: mIoU scores for BlackFed v1 and v2 in comparison with individual and FL-based training strategies for Cityscapes. "Local" represents test data from the center. "OOD" represents mean mIoU on test data from rest of the centers. For FedAvg and Combined Training, just one model is trained. Hence, its performance is noted only in each of the local test datasets.

| | C10 | | C11 | | C12 | | C13 | | C14 | | C15 | | C16 | | C17 | | C18 | |
|---|---|---|---|---|---|---|---|---|---|---|---|---|---|---|---|---|---|---|
| Method | Local | OOD | Local | OOD | Local | OOD | Local | OOD | Local | OOD | Local | OOD | Local | OOD | Local | OOD | Local | OOD |
| Individual | 0.49 | 0.45 | 0.62 | 0.57 | 0.58 | 0.54 | 0.38 | 0.39 | 0.41 | 0.39 | 0.56 | 0.50 | 0.38 | 0.39 | 0.66 | 0.55 | 0.37 | 0.39 |
| BlackFed v1 | 0.69 | 0.71 | 0.74 | 0.70 | 0.68 | 0.71 | 0.68 | 0.71 | 0.74 | 0.69 | 0.73 | 0.68 | 0.65 | 0.66 | 0.69 | 0.70 | 0.72 | 0.70 |
| BlackFed v2 | **0.72** | **0.73** | **0.77** | **0.73** | **0.71** | **0.73** | **0.72** | **0.74** | **0.78** | **0.73** | **0.78** | **0.74** | **0.74** | **0.73** | **0.80** | **0.73** | **0.76** | **0.74** |
| Combined Training | 0.74 | - | 0.80 | - | 0.74 | - | 0.73 | - | 0.81 | - | 0.79 | - | 0.77 | - | 0.82 | - | 0.78 | - |
| White-box Training | 0.73 | 0.76 | 0.79 | 0.75 | 0.72 | 0.75 | 0.71 | 0.76 | 0.78 | 0.75 | 0.79 | 0.75 | 0.72 | 0.74 | 0.81 | 0.75 | 0.78 | 0.75 |
| FedAvg [34] | 0.76 | - | 0.80 | - | 0.75 | - | 0.74 | - | 0.82 | - | 0.83 | - | 0.77 | - | 0.84 | - | 0.81 | - |

