# OpenReview forum: "Federated Black-Box Adaptation for Semantic Segmentation"
_NeurIPS.cc/2024/Conference — NeurIPS 2024 poster_

### Official Review · Reviewer_XV1b · 2024-06-23

**Soundness:** 3
**Presentation:** 3
**Contribution:** 2
**Rating:** 5
**Confidence:** 4

**Summary:**

The authors proposed a black-box tuning method to address the problem of Semantic Segmentation in FL. Specifically, they propose to split the network into two parts and store them in server and clients separately. The server modules are optimized via first-order optimization while the client modules are optimized via black-box optimization. The proposed method achieves promising results on different benchmarks.

**Strengths:**

1. The writting is clear and the presentation of the method is easy to follow.
2. The proposed method of combining split learning and black-box optimization is interesting.
3. The experiments are thorough and the results of the method are promising.

**Weaknesses:**

1. From my understanding, the privacy problems still exist. Since the authors claimed that they are using a two-layer CNN at the client, which is relatively light-weight and compact compared to the network used at server, the attackers at server may still be possible to reproduce the training data using the intermediate output (the uploaded values from each client). The authors may need to add further discussion about this issue.
2. In the process 1b (Figure 2), how is the first-order gradient computed at the server? Are the ground-truth semantic maps uploaded to the central server? Or the network predictions will be downloaded to different clients and then evaluated? For the first solution, there will be privacy threatens. For the second one, the communication costs will be very large. The authors should clarify this in more details.
3. The authors introduce a new two-layer CNN at each client, could the network DeepLabv3 (the network at central server) directly be split in the clients and server?
4. It would be interesting to compare with other FL algorithms which also applies Black-box Tuning such as [1].

[1] Sun J, Xu Z, Yin H, et al. Fedbpt: Efficient federated black-box prompt tuning for large language models[J]. arXiv preprint arXiv:2310.01467, 2023.

**Questions:**

Please refer to the questions in the Weaknesses section.

**Limitations:**

The authors did not address the limitations of the proposed method.

---

> ### Author Rebuttal · Authors · 2024-08-05
>
> We thank the reviewer for the valuable comments.
> 1) Reproduction of training data:
>
>     a) The clients can arbitrarily choose their networks. In the current setting, there is no dependence between the clients or the client and the server other than the output size of the client's last layer. The current work proposed a two-layer CNN as an example network.
>
>     b) Even for a two-layer network, lets say an attacker knows the intermediate representations. Then the task for the attacker would be to find the input x and the function f given y where f is a small network and y is the leaked intermediate output. Now, to find out x, one could solve the equations y = f(x) and y’ = f’(x), where f’ denotes the derivative with respect to x. However, this method requires gradients to be known as well, which is not the case with out method.
>
>     c) Furthermore, one method could be to learn a new network to produce the same outputs as the client models, given some input. In this case, the attacker would initialize a new two layer CNN and would have to query the original client model with learnable inputs. This way, they could train the new model with the client outputs as ground truths. However, for this case as well, the attacker would require access to the actual client models.
>
> 2) Sending Masks to the Sever: In the proposed work, masks are sent to the server for FOO. Even with this information, the attacker would require raw data to generate a new model that can emulate the client without gradients. For example, consider stable diffusion or similar methods that can recreate images given their masks. These would be based on public datasets and hence, would not be able to generate PII information that will be present in the raw data. In fact, given the mask, they can still generate synthetic data, but that cannot be considered as replicating raw data since it would still use a distribution of pixels similar to public datasets and not the private client data. We believe that not requiring model information transfer and gradient transfer is an important step in the direction of better privacy-preserving FL over existing methods.
>
> 3) Directly Splitting the Server Model:  The network at server could also be split into two parts as mentioned by the reviewer. While this is a valid strategy, this would limit all the clients to use the same network. At the same time, the server would potentially have the complete model architecture information, which is not required in the current setting. The motivation behind proposing a new lightweight client model was that this would allow different clients to have their own design choices for the model with the condition that the output should have a particular dimension. Since the client is updated using ZOO, having a lightweight model might also help in getting a better performance.
>
> 4) FedBPT as added black-box method: We thank the reviewer for pointing out the FedBPT method. We will add it in the related work section. Please note that FedBPT was proposed to work with foundation models and as a prompting mechanism. Hence, it uses CMA-ES, a ZOO method to learn a prompt for the pretrained foundation model at each client and aggregates these prompts at the server. Since we want to learn the network parameters themselves and not the prompts, we cannot directly compare with this method. Instead, for the experiments, we tried replacing SPSA-GC with the CMA-ES method for updating the client in our algorithm. However, we see that it kills the program due to the large number of parameters in the network as compared to prompts. CMA-ES is an evolutionary algorithm that maintains a big group of candidate values for each trainable parameter, that is perturbed towards a better loss value in each iteration. Since the number of parameters is very large, we were not able to run CMA-ES as the ZOO algorithm.

---

> ### Author Response · Authors · 2024-08-10
>
> Dear reviewer,
>
> Thank you for the comments on our paper.
>
> We have submitted the response to your comments and a PDF file. Please let us know if you have additional questions so that we can address them during the discussion period. We hope that you can consider raising the score after we address all the issues.
>
> Thank you

---

> > ### Comment · Reviewer_XV1b · 2024-08-11
> >
> > During the rebuttal, the authors have addressed most of my concerns. Therefore, I would increase my score to Broadline Accept.

---

### Official Review · Reviewer_ykrj · 2024-07-11

**Soundness:** 2
**Presentation:** 3
**Contribution:** 2
**Rating:** 5
**Confidence:** 3

**Summary:**

This manuscript introduces a federated learning framework for semantic segmentation that neither requires knowledge of the model architecture nor involves transferring gradients, thereby preventing privacy leakage. BlackFed incorporates split neural networks and first/zero order optimization for training the server and clients, respectively.

**Strengths:**

(1) The writing of the paper is clear.

(2) The concept presented in this manuscript is interesting.

**Weaknesses:**

(1)	In addressing privacy leakage caused by sharing weights and gradients in current federated learning methods, this manuscript proposes BlackFed, which iteratively updates client and server networks. However, this approach involves transmitting extracted features and labels between the client and server, leading to privacy leakage and security risks. For instance, some algorithms like stable diffusion [1] can reconstruct client images from these features and segmentation masks. Therefore, I believe BlackFed does not effectively protect privacy and fails to show superiority over methods based on network weights and gradients.

(2)	There are too few comparison methods. In the Related Work section, many federated learning methods for segmentation tasks are introduced; FedSeg and FedSM could be used for comparison.

(3)	BlackFed preprocesses client data using a two-layer convolutional neural network on clients, allowing different clients to extract global features. These preprocessed features are then sent to the server for segmentation. The client-side training follows the method proposed in [2], which limits the approach's innovation.

(4)	In the inference phase, the features extracted by the client network need to be uploaded to the server each time. Does this significantly increase testing time consumption?

(5)	FedPer [3] is another federated learning method using split neural networks. It is recommended to introduce it in the Related Work section and add comparison results in the experimental section.

[1] High-Resolution Image Synthesis with Latent Diffusion Models

[2] Blackvip: Black-box visual prompting for robust transfer learning.

[3] Federated Learning with Personalization Layers

**Questions:**

(1) More comparison of Federated learning methods for segmentation should be performed;

(2) All experiments were conducted using DeepLabV3 as the server segmentation network. It is recommended to include experimental results on different segmentation architectures.

**Limitations:**

The motivation behind BlackFed is privacy protection. However, the feature maps extracted by clients and segmentation masks are also transmitted to the server, making them vulnerable to attacks that could reconstruct the images from different clients. Therefore, this method does not achieve the goal of privacy protection.

---

> ### Author Rebuttal · Authors · 2024-08-05
>
> We thank the reviewer for the valuable comments.
>
> 1) Reproduction of Training data: As the reviewer suggested, the intermediate representations and segmentation masks can be used along with diffusion models. However, to train these models, one would still require the raw data to act as ground truth labels during training. If one were to use public datasets for training such diffusion models, the diffusion model would still learn to generate images from a similar distribution to the public datasets. This is not the same as regenerating raw data since raw data can have personally identifiable information (PII) that won't be learnt by diffusion models which are trained on public datasets. In such scenarios, in fact, the output from Stable Diffusion can be considered as a good source for synthetic data, which looks similar to raw data but would arise from the public data distribution. In fact, some recent works also highlight the property of stable diffusion and other image generation methods to copy the data they were trained on [1-4]. In such a scenario, in the absence of raw data from the client, the best a diffusion model would be able to do is to generate images from public datasets similar to the segmentation mask, which we think cannot be considered as reproduction of client data. We believe that not requiring model information transfer and gradient transfer is an important step in the direction of better privacy-preserving FL over existing methods.
>
> [1] Frame by Familiar Frame: Understanding Replication in Video Diffusion Models
>
> [2] Diffusion Art or Digital Forgery? Investigating Data Replication in Diffusion Models
>
> [3] Analyzing bias in diffusion-based face generation models
>
> [4] Replication in Visual Diffusion Models: A Survey and Outlook
>
> 2) Adding more Comparisons: FedSeg is a method that is built upon FedAvg and works by sharing model weights across server and client. Hence, this would act as an upper bound for our method, just like FedAvg. We verify this for CAMVID and Cityscapes and add it to Table 1 presented in the extra table page for rebuttal. We see that for both the datasets, it generally performs better than FedAvg. Please note that this cannot be directly compared with BlackFed since our approach operates in a more restricted setting where model sharing is not allowed. We also add other methods like FedPer in this table, as suggested.
>
> 3) Novelty for ZOO: Please note that we do not claim novelty for the ZOO method used. In fact, we cited the BlackVIP paper as being the work that introduced SPSA-GC as a ZOO method (ref. No. 38 in the paper). The major contributions of our work include formulation of the FL problem in the blackbox setting using split networks to allow gradient-free updates. The proposed approach formulates the FL problem using a lightweight client and a parameter-heavy server, along with a round-robin algorithm that can allow SPSA-GC to work well, since it was originally meant to fine-tune pre-initialized foundation models. The proposed approach shows that by alternating between clients, and updating them with ZOO and FOO, one can achieve good performance. In addition, we identify the catastrophic forgetting problem and come up with a solution using hashmaps.
>
> 4) Inference: During inference, uploading the features to server would increase time as pointed out. Some ways to reduce this would be to upload batches of features at the same time that can reduce the amortized time cost. An alternative is to get a copy of the server initialized with the hashmap weights for the client after training is complete. Then, client can perform the inference locally. This assumes that the client has sufficient compute power. Note that the entire hashmap need not be shared, but only the entry corresponding to the client.
>
> 5) Additional Method FedPer: We thank the reviewer for pointing out this method. We will add it to the related work and comparisons in the revised paper. In addition, we added results with FedPer in Table 1 in the extra rebuttal page. However, FedPer does not use split networks. Instead, it is similar to FedAvg, the difference being that not all weights are shared with the server. Here, the majority of the weights are shared with the server and aggregated. The remaining weights are "personal" to the model and allow for better local performance. These are not shared with the server. In contrast, there are no weights shared between the client and server in Blackfed. The output from the client is given to the server, where it is further processed, making this a split network. Furthermore, The FedPer paper only performs the task of classification. In order to adapt this for segmentation, we considered the weights of the classifier head of DeepLab v3 as personal weights and the rest of the backbone weights were averaged in the server. We find that in most cases, this improved local performance over FedAvg, but reduced OOD performance.
>
> 6) Ablations with more architectures: Table 3 in the paper compares performance with three different architectures for the server. This includes Unext and Segformer in addition to DeepLab. Segformer is a transformer-based method while the others are CNN-based.

---

> > ### Comment · Reviewer_ykrj · 2024-08-12
> >
> > Thanks for the author's rebuttal. The majority of my concerns have been addressed. However, the explanation of BlackFed's privacy protection is not convincing to me, so I will keep my rating unchanged.

---

> > > ### Author Response · Authors · 2024-08-12
> > >
> > > We are happy that we could address most of your concerns. For the privacy protection concern, please note that we do not claim a perfectly attack-proof method in this work. In lines 42-52 of the main paper, we reference works that propose attacks using the existing FL frameworks. Hence, we propose a mechanism which would not satisfy the necessary conditions of gradient transfer / model transfer for these attacks. In the contributions section from lines 53-60, we clearly state that we propose a new framework for FL without gradient and model transfer, and do not claim that it would solve the privacy preserving problem completely.
> > >
> > > However, we will be adding a discussion on this in the future work section that can encourage more research on attacks and defenses given the new framework. We believe that our work is an important step in the direction of FL algorithms which involve minimal transfer and it will encourage future research in this direction, diffusion-based attacks and defense being one such example.
> > >
> > > We hope that you would consider raising the score if we address the present concern.
> > > Thanks

---

> > > > ### Comment · Reviewer_ykrj · 2024-08-13
> > > >
> > > > Thanks for the author's comment. There is indeed no perfect federated method that can achieve perfect privacy protection. Since the main contribution of this paper lies in privacy protection, it would be beneficial to substantiate the privacy protection performance through experiments or mathematical analysis in the next version. Finally, I will increase my score to Broadline Accept.

---

> ### Author Response · Authors · 2024-08-10
>
> Dear reviewer,
>
> Thank you for the comments on our paper.
>
> We have submitted the response to your comments and a PDF file. Please let us know if you have additional questions so that we can address them during the discussion period. We hope that you can consider raising the score after we address all the issues.
>
> Thank you

---

### Official Review · Reviewer_Ftdr · 2024-07-12

**Soundness:** 3
**Presentation:** 3
**Contribution:** 2
**Rating:** 4
**Confidence:** 3

**Summary:**

In this work, the authors introduce BlackFed, an FL algorithm that enables distributed learning without transfer of gradients or model weights. This characteristic distinguishes the approach as a black-box model compared to existing FL methods, which can be considered as white-box approaches. Recent research on attacking FL methods require the knowledge of either the gradients or the model information, thus rendering BlackFed more resistant to such attacks since gradients or weights are not shared. BlackFed consists of a global server and multiple clients, each possessing their own data. The server is using first order optimization while the client weights are updated using zero order optimization in a round-robin fashion. This introduces the effect of catastrophic forgetting in the network, for which the authors propose a simple hashmap-based approach. The white-box methods, despite being a black-box method itself. Extensive experimentation on the natural and medical domain datasets highlights the effectiveness of BlackFed.

**Strengths:**

In this work, a new approach, named BlackFed, is proposed. For segmentation, it uses FL that does not involve gradient transfer between the server and the client and at the same time, it passes no knowledge about the client model architecture to the server, thereby avoiding the necessary conditions for these attacks.
The strengths is as follows:
1. BlackFed - a black-box algorithm that facilitates distributed learning for semantic segmentation without transferring model information or gradients between the client and the server is proposed. The authors formulate the FL problem using split-nn and use first and zero order optimization for training the server and the clients, respectively.
2. To reduce the effect of catastrophic forgetting, the authors propose a simple additional step during training. After updating the server weights for a given client during training, the updated weights of the server model are stored in a hashmap indexed by the index of the client. During inference for a given client, the latest weights of the client model and the indexed weights of the server model to perform the forward pass are used.
3. The proposed approach is evaluated on four segmentation datasets and it shows it’s effective as a distributed learning method by showing improvements over individual training.

**Weaknesses:**

The weakness is as follows
1. For Table 1, it's indeed the proposed methods work well. But there is not experimental result for deeplabv3 on a single machine. Deeplab v3 achieves 80.0 on Cityscape val dataset on https://paperswithcode.com/lib/detectron2/deeplabv3-1. The paper generates 18 clients for Cityscapes. What's the relationship between the Table 1 evaluation result and benchmark result on Cityscape? This may make the paper not convincing enough.
2. It's unclear why "As the model complexity increases from UNext to DeepLab to Segformer, we observe a decrease in individual training performance. ".
3. The image size for Cityscape is  256 × 512. why using so small resolution? it needs to do more experiments on higher resolution.
4. What's the details for " Consequently, we start the DeepLabv3 network in the server from the second layer, which expects a 64-channel input. " ? Does it mean the deeplabv3 trained on server only unfreezed from the second layer? so how about the client?

**Questions:**

No. Please see Weakness.

**Limitations:**

No. Please see Weakness.

---

> ### Author Rebuttal · Authors · 2024-08-05
>
> We thank the reviewer for the valuable comments.
> 1) Comparison with the baseline from papers with code: The dataset of Cityscapes has data from 18 different centers. In the original splits of Cityscapes, the training data contains images from some of these centers, while the data from rest of the centers is in the validation and testing sets. In our case, we want each center to have its own training, validation, and testing data. The goal of the Federated Learning model would be to do well on test data from other centers in addition to doing well on the in-house test set, indicated by the “OOD” and “Local” columns in the tables. Hence, the data splits are done differently. Training DeepLab v3 on single machine is the same as “Combined training” in Table 1, where all the training data is merged to train a single model, that achieves a DSC of 0.77, which is close to the number shown on PapersWithCode. This represents an upper bound to what BlackFed can achieve since training on a single machine means that it has access to data from all centers.
>
> 2) Clarification on Trend of Model Complexity vs Performance: The model complexity includes the number of trainable parameters which increase from UNext to DeepLab to Segformer. But in table 3, row 1, we see that the individual performance increased for DeepLab but decreased drastically for Segformer, even though it has the most number of parameters. Since the data is limited in this case, we see that increasing the number of trainable parameters beyond a certain point causes the validation performance to suffer. However, for the case of combined training, the data quantity is higher, so val performance increases from UNext to DeepLab v3 to Segformer as expected. This is the desired trend in the FL algorithm as well since it also has access to more data than the limited data case of individual training. This trend is seen in BlackFed as the performance increases from Unext to DeepLab and does not drop when the model complexity increases from DeepLab to Segformer.
>
> 3) Reason for Lower Resolution: We wanted to do the training of a given client on a single GPU, since institutes like medical centers do not have a high compute power. Hence, we downscale the image without affecting its aspect ratio. Training on a larger resolution would require much higher computation. Since we are emulating all client and server computations in the same compute center, this would require increased compute, especially in the case of 18 centers like Cityscapes. However, we plan to release a model zoo on Github with pretrained checkpoints at different resolutions since training with larger resolutions can improve performance.
>
> 4) Clarification on Server Architecture: All the parameters in the approach are trainable and there are no frozen layers. In our experimental setup, we use a two-layer Convolution Net in the client to reduce the load on each client. The output of this layer is a feature map of the shape H X W X 64. This is transmitted to the server for further processing. Hence, the first layer of the server architecture should expect an input with this shape. We are using DeepLab v3 architecture for the server. However, instead of starting from the Conv1 layer in DeepLab, we are starting from Conv2 (and deleting Conv 1 from the network), which also expects an input with 64 channels. Hence, in order to continue the training in the server from the client's output, we designed the server architecture in such a way. In other words, this can be considered as one network, the first two layers of which are in the client while the rest of it is in the server, hence the term “split network”.

---

> ### Author Response · Authors · 2024-08-10
>
> Dear reviewer,
>
> Thank you for the comments on our paper.
>
> We have submitted the response to your comments and a PDF file. Please let us know if you have additional questions so that we can address them during the discussion period. We hope that you can consider raising the score after we address all the issues.
>
> Thank you

---

> > ### Author Response · Authors · 2024-08-12
> >
> > Since we are approaching the end of the discussion period, this is a gentle reminder. Please let us know if you have additional questions so that we can address them in the next day during the discussion period.
> >
> > We hope you would consider raising the score if we have addressed your concerns satisfactory
> >
> > Thanks

---

> > > ### Author Response · Authors · 2024-08-13
> > > **Reminder for the discussion**
> > >
> > > Dear reviewer,
> > > Today is the last day for the discussion period. Please let us know of we were able to address all the concerns regarding the paper. If so, kindly consider raising the score.

---

### Official Review · Reviewer_zJW7 · 2024-07-12

**Soundness:** 3
**Presentation:** 2
**Contribution:** 4
**Rating:** 7
**Confidence:** 3

**Summary:**

A common issue now with federated learning systems is that they are not completely private owing to gradients transferred between the clients and the global server during training or by knowing the model architecture at the client end.

The paper proposed a workaround to this by removing the need for the passage of the gradients from client to server or knowing the model at the end. For the gradients, the authors propose zero-order optimization (ZOO) to update the client model weights and first-order optimization (FOO) to update the server weights.

While achieving the above, they also perform reasonably well on most datasets.

**Strengths:**

1. An important direction of research where the federated framework does not require gradient transfer from client to server and an approach to tackle catastrophic forgetting in that framework.
2. Comparable results to gradient accessed methods across all datasets.
3. Significant reduction in training costs as compared to previous methods.

**Weaknesses:**

1. The case of polypgen (given the dataset size, medical background, and criteria of data collection) is where there is the most requirement/use of such a setup. And it underperforms over there. "This behavior may be related to the data distribution of Polypgen and suggests that BlackFedv2 is not able to correctly avoid the catastrophic forgetting for centers C5 and C6. ". --- I think this has less to do with catastrophic forgetting and more to do with the model just not able learn. The data points are very few across all the centers. Given how the latest weights are stored, the weights used are just poor as they are not able to learn appropriately given the tough case of low data centers. It may also be that it is overparameterised by the other centres. If it was the case of it being affected by not being able to solve catastrophic forgetting, this would have been reflected everywhere. Especially in C1 C2 of CAMVID.

**Questions:**

1. Why was the white-box training so subpar in ISIC? Further, while I am not well-versed in the literature, I imagine there are ways to tackle catastrophic forgetting in the white-box method as well. Would it be possible to do an ablation with that? Comparing V1 and white box, apart from ISIC, it beats V1 in all cases.

2. Given that CAMVID has a Significantly lesser number of classes and centers, why is the performance better/on-par (and more stable) in the Cityscape dataset for BlackFed but worse in the case of "individual"? Even considering the majority classes.

**Limitations:**

1. "After updating the server weights for a given client during training, we store the updated weights of the server model in a hashmap".  - This step may increase significantly with scale, especially with multiple client updates. Lower floating point operations are indeed handy, but this is coming at the cost of high memory requirements.

2. Issues with performance in the most important setting.

Suggestion -

3. I think in Table 1 and 2, Bolding the highest value and underlining the highest value without gradients will be a more proper representation.

---

> ### Author Rebuttal · Authors · 2024-08-05
>
> We thank the reviewer for the valuable comments.
> 1) Low Performance on Polypgen: This case indeed serves as an important application area. Here, for 4 out of 6 centers (C1, C2, C4 and C5), it is beneficial to use the FL method. But for C3 and C6, the OOD performance decreases. This could be due to other centers dominating the training, thus causing catastrophic forgetting for particular centers like C6. But as pointed out by the reviewer, this should have been a problem elsewhere as well. In such cases, maybe the learnt weights are not good due to a more adverse distribution shift between centers in Polypgen as compared to CAMVID. In such cases, it would be important to control the training of the FL algorithm to make it more robust, which is an interesting and important direction for future research in this area.
>
> 2) Counter-Intuitive Performance of Whitebox on ISIC: As pointed out by the reviewer, for ISIC, the whitebox method strangely performs subpar. We verified this through various runs and changes to the hyperparameters. This anomalous behaviour may be due to the model getting stuck in some local minima during training because of overparametrization.
>
> 3) Comparing Individual Performance for CAMVID and Cityscapes: In the Blackfed case, for both datasets, the server model benefits from the entire dataset, hence the simliar performance. However, in the individual case, the model performs subpar in some of the clients. For CAMVID, individual performance of C1 is much lower than others. But for Cityscapes, there are some clients with a good performance and some clients with subpar performance. Hence, since the number of clients is more, on an average, the indivual performance is lower. This also depends on the choice of model architecture and data distribution since the number of data points is so less.
>
> 4) Additional Memory Overhead: As the reviewer pointed out, the proposed approach comes with an added memory overhead for storing the hashmap, which scales with the number of clients. However, the motivation for the method was based on the assumption that while the client has limited memory, the common server can have a large memory. Based on the model architecture, each entry in the hashmap would be of the order of 100 MB or less. If there are 100 clients, this would translate to 10 GB, which is a realistic number. For larger models, we believe that the memory overhead cost would not be a bottleneck in comparison to the cost of training such models. One interesting direction in such cases would be using Parameter Efficient Finetuning Methods (PEFT) like adapters or LoRA and only storing them.

---

> ### Author Response · Authors · 2024-08-10
>
> Dear reviewer,
>
> Thank you for the comments on our paper.
>
> We have submitted the response to your comments and a PDF file. Please let us know if you have additional questions so that we can address them during the discussion period. We hope that you can consider raising the score after we address all the issues.
>
> Thank you

---

> > ### Comment · Reviewer_zJW7 · 2024-08-11
> >
> > I would chose not to change my decision (for better) based on the fact that - 1. Thinking of models in terms of 100MB is not fair. No production model is going to be that small.
> >
> > 2. “ Hence, since the number of clients is more, on an average, the individual performance is lower.” - This is a fundamental  problem which again is not being addressed in any form.
> >
> > The results overall are quite normal. However, given the novelty to form a gradient-free approach is why I proposed the accept and that is the only contributing factor.
> >
> > Thank you for the detailed rebuttal.

---

### Author Rebuttal · Authors · 2024-08-05

We thank all the reviewers for their valuable comments. We tried to address all concerns in the respective rebuttal sections. Here, we would like to write the global rebuttal for two common concerns raised by the reviewers:

1) More comparison experiments: We added two new results in Table 1 of the paper (shown in the attached pdf file) for the suggested methods FedPer [1] and FedSeg [2]. Both of these methods involve model weights transfer similar to FedAvg and so can be considered as upper bounds for our method.

    (a) In the case of FedPer, the majority of the weights are shared with the server and a small head is retained at the client to allow for more personalization of client models. We see that this improves local performance but is not able to perform as well on OOD data. While this is defined for classification, for comparison, we adapt it for segmentation by keeping the final classifier head of our network 'personal' to the model and don't share its weights with the server. We find that our method comes on par with this method on OOD distribution, despite being a black box method not involving model weight transfer.

    (b) In the case of FedSeg, all the weights are shared with the server similar to FedAvg. During the training of the client, an additional step is taken to align the behaviour of the model for OOD and local distributions better. This method was introduced as an improvement to FedSeg. Our method performs on par with FedSeg without sharing model weights.

    (c) One of the reviewers suggested a comparison with FedBPT [3], which uses the ZOO method called CMA-ES for learning a prompt to foundation models. This prompt is shared between the client and the server while the same frozen foundation model is used at all clients and the server. CMA-ES is an evolutionary method and hence, requires lots of candidates for each of the learnable parameters. Over several iterations, these candidates converge to an optimal value for the parameter. In FedBPT, the learnable prompt is very tiny compared to the client part of the network in our case in terms of learnable parameters. Hence, when we used CMA-ES instead of SPSA-GC as the zero-order optimization method, it always crashed the program. Hence, we believe that CMA-ES would not be a feasible solution for optimizing a larger number of parameters.

[1] Federated Learning with Personalization Layers

[2] FedSeg: Class-Heterogeneous Federated Learning for Semantic Segmentation

[3] Fedbpt: Efficient federated black-box prompt tuning for large language models

2) Discussion on regeneration of training data: The reviewers had concerns about whether attackers could target the method with existing methods like Diffusion Models or other mechanisms. We present our views on this topic below:

    (a) For attacking methods that generate new models to approximate raw data with the lowest error, an optimization problem is solved. Here, the goal would be to find input x that minimizes the error between the predicted value of the original model and the predicted value of the new model. However, these methods would either require the gradients or the ability to query the original client model multiple times, both of which are not available in our proposed method.

    (b) Diffusion models present a more interesting challenge since methods like Stable Diffusion can generate synthetic images based on the shared mask and labels. However, various works have found that these Image Generation methods mimic the data distribution that they are trained on [1-4]. Hence, even if such methods are used to attack BlackFed, they would generate data with pixel distribution from existing public datasets and lack the Personally Identifiable Information (PII) in the raw client data, which is private and not accessible to the diffusion model for training. In such a scenario, while the generated synthetic images would be valid data, they should not be considered as replicating the raw input data.

[1] Frame by Familiar Frame: Understanding Replication in Video Diffusion Models

[2] Diffusion Art or Digital Forgery? Investigating Data Replication in Diffusion Models

[3] Analyzing bias in diffusion-based face generation models

[4] Replication in Visual Diffusion Models: A Survey and Outlook

---

### Comment · Area_Chair_77oX · 2024-08-09
**Rebuttal is online - please respond**

Dear Reviewers,

Authors carefully prepared their rebuttal - trying to address the concerns you have raised. Please check the rebuttals and join the discussion about the paper.

Regards,

---

### Decision · Program_Chairs · 2024-09-25

**Decision:**

Accept (poster)

**Comment:**

Authors present a gradient-free federated learning approach in which knowledge of the client architecture nor the weights are not required by the server. The client and server networks are optimized differently. The main claim for the article is that this approach is more privacy preserving because the gradients and weights of clients remain at the client. However, representations and masks need to be transmitted to the server which may still cause some privacy issues.

The main concerns for the rebuttal were on experimental analysis and the privacy concerns. In the experimental analysis, more comparisons and clarifications were required. For the latter, transmitting representations and masks seem to violate the privacy concerns. In other words, the proposed technique also cannot guarantee full privacy of the data.

During the rebuttal authors successfully answered questions on experimental analysis. Questions on the privacy concerns remained open to the best of my understanding. Among four reviewers, three of them suggested to accept the article, two with BA and one with A. Only one reviewer suggested BR. However, I believe concerns this reviewer raised were addressed to some extent during the rebuttal. Unfortunately, the reviewer did not revise their score.

Given the positive reviews, acknowledged technical novelty, successful rebuttal and the final ratings, I believe this article would be a good contribution to the conference.